# Main Aspects of Preparing Diabetic Patients in Poland for Self-Care

**DOI:** 10.3390/ijerph191811365

**Published:** 2022-09-09

**Authors:** Agnieszka Pluta, Alicja Marzec, Edyta Kobus, Beata Sulikowska

**Affiliations:** 1Department of Preventive Nursing, Faculty of Health Sciences, Ludwik Rydygier Collegium Medicum, Nicolaus Copernicus University, 87-100 Toruń, Poland; 2Tadeusz Borowicz Provincial Infectious Diseases Hospital in Bydgoszcz, 85-681 Bydgoszcz, Poland; 3Department of Nephrology, Hypertension and Internal Diseases, Faculty of Medicine, Ludwik Rydygier Collegium Medicum, Nicolaus Copernicus University, 87-100 Toruń, Poland

**Keywords:** diabetes, patient, education, self-care, self-observation, nurse

## Abstract

Diabetes is a lifestyle disease which can cause many complications and organ-related disorders. The aim of the study was to analyze selected aspects of preparing patients with diabetes for self-care. The study group consisted of 190 people diagnosed with type 1 and type 2 diabetes, including 101 women and 89 men. The mean age of the respondents was 42.2 ± 13.4 years. The study was conducted using an anonymous self-designed questionnaire containing 50 questions. Among the respondents, 23.2% did not control their glucose levels at home. The respondents most often measured glucose once a day (33.6%) or three times a day (26.7%). A total of 64.7% of the respondents declared that they kept a self-monitoring diary. The knowledge of the symptoms of hypoglycemia and the ability to properly manage it was declared by 64.8% of the respondents. A total of 52.1% of the patients did not undertake any activity lasting more than 30 min at least 3 times a week, and 75.2% described their condition as very good and good. Independent participation in therapy, i.e., taking hypoglycemic drugs or insulin, was declared by 63.7% of the respondents. Despite undergoing therapeutic education, the study population diagnosed with diabetes still shows deficiencies in terms of awareness of proper health behaviors. Objective results showed that the patients had insufficient knowledge and skills in terms of self-care and self-observation, blood glucose and blood pressure measurements, physical activity, diet therapy as well as adherence to pharmacotherapy recommendations. Despite the good general preparation for self-care as declared by the respondents, these patients require further systematic, individual educational activities. The results of the present study have implications for nursing practice, patient therapeutic education, and the functioning of the public health and healthcare systems. The number of diabetic patients is constantly increasing. Patients require coordinated care and individualized therapeutic education in order to be prepared for self-care and self-management, thus reducing the risk of complications. Delaying the occurrence of potential complications provides patients with a chance to live an active private and professional life, and protects the health care system from carrying the cost burden of expensive highly specialized services.

## 1. Introduction

According to the World Health Organization, diabetes is a non-communicable lifestyle disease. Data provided by the Organization for Economic Co-operation and Development (OECD) and the European Commission shows that the number of diabetics worldwide is gradually increasing. In 2014, this diagnosis concerned 422 million adults worldwide. The forecast for 2035 predicts a considerable increase in the incidence of diabetes—591.9 million cases. The highest incidence rates are observed in highly developed countries, in both Americas and Europe. In 2014, there were 64 million adult diabetics in Europe, while for Poland the latest data indicate about 3 million cases [1,2,3].

Diabetes mellitus is a chronic, progressive disease that carries many complications and organ burdens. Clinical experts have developed a consensus on the care of patients with diabetes. The standard of patient care is recommended for the diagnosis of diabetes, pharmacological and dietary treatment, as well as early and late prevention [4,5]. It is consistent with the guidelines adopted by the Polish Diabetes Society [6].

Diabetes care requires appropriate competencies to be demonstrated by the entire therapeutic team. It includes several interrelated components: pharmacotherapy, behavioral therapy (diet, exercise), specialist consultations to assess metabolic control and the severity of late complications, education in lifestyle modification, and psychological support. Care should be patient centered, taking into account their individual circumstances, abilities, needs and preferences. It is also necessary for specialists from related fields to cooperate as part of coordinated care, due to the multidirectional nature of diabetes complications and comorbidities [6,7,8].

Pursuant to the legal regulations in Poland, a nurse is entitled to independently perform preventive services without a doctor’s order, including the education of diabetics and their families, if they have completed a specialist or qualification course, or if they have the title of a specialist in the field of nursing, or they hold a master’s degree in nursing [9].

In Poland, diabetics benefit from free-of-charge specialist services in the field of diabetes in outpatient specialist care (AOS) clinics, the activities of which are financed from public funds. As part of this form of care, there is a service called nursing advice. The scope of the nurse’s services as part of diabetes counseling includes:(1)Health education and health promotion;(2)Selecting methods of treating wounds as part of medical services provided by a nurse independently without a doctor’s order;(3)Prescribing drugs containing specific active substances, including prescribing them, with the exception of medicines containing very strong substances, intoxicants and psychotropic substances;(4)Issuing a prescription for drugs prescribed by a doctor, as part of the continuation of treatment, with the exception of medicines containing very potent substances, intoxicants and psychotropic substances;(5)Prescribing certain medical devices, including issuing of an orders or prescriptions for them;(6)Issuing a referral for specific diagnostic tests, with the exception of tests requiring diagnostic and treatment methods that pose an increased risk for the patient [10].

The first consultation of a diabetic patient is carried out in cooperation between a diabetologist and a nurse, and the continuation of treatment may be carried out independently by the nurse on the basis of written information issued by a diabetologist.

In the treatment of diabetes, the initial education (at the diagnosis of the disease) of a person with diabetes treated with a diet or diet and oral antihyperglycemic drugs lasts 5 h, while the education of a patient treated with insulin lasts 9 h, and a person treated with a personal insulin pump and glycemic monitoring systems receives 15 h of education in an outpatient or inpatient setting, depending on the situation of the person with diabetes and the possibilities of the care facility. Every person with diabetes should have diabetes education initiated as soon as possible after the diagnosis of the disease. Diabetes education is continuous. The Polish Diabetes Association recommends periodic (annual) checking of the patient’s knowledge, which could be performed in person or in electronic form using telecommunications techniques. Subsequent screening and re-education are performed when new risk factors/complications emerge. In addition to individualized education, group educational programs (6–10-person groups) are implemented. Education is conducted by properly trained people (doctors, diabetes educators, nurses, nutritionists) [11].

Any educational program, regardless of the type of diabetes, should therefore include:-Familiarizing the patient with the causes, symptoms and course of the disease;-Teaching the patient to recognize the symptoms of hypoglycemia, hyperglycemia, prevent and how to manage them;-Showing the importance of self-control in diabetes through glycemic control, glucose control, blood pressure control, diary keeping and weight measurement;-Prevention of long-term complications by constant monitoring of parameters such as fundus examination, blood tests, ECG, foot examination, and assessment of kidney function;-Presentation of the role of physical effort;-Teaching the rules of nutrition;-Informing about the rules of insulin storage;-Teaching the technique of preparing an injection, how to perform it, showing the injection sites [12,13].

In modern medicine, the patient is the subject of care. He is expected to actively participate in therapy, presenting a partnership attitude towards the therapeutic team, which is associated with joint decision making and joint responsibility for the effects of therapy. In modern nursing, the care of the chronically ill focuses on preparing the patient for living with the disease, for self-management and self-care—as defined by Dorothea Orem’s theory—with emphasis on personal responsibility for one’s health [14,15]. Bearing in mind the assumptions of Dorothy Orem’s theory about the systems of cooperation with the patient proposed by her: compensatory, partially compensatory, and supportive-educative, wherein the nurse should define the patient’s abilities, expectations, and environmental conditions. The above elements are necessary to choose the form of work with the patient [14,15]. A patient with a poor clinical condition, who is physically unfit, will require a lot of involvement from the nurse. In contrast, a fit patient with high intellectual potential, open perception, and motivation will only require guidance and support.

Contemporary models of care for patients with chronic diseases such as diabetes focus on preparing them for self-care. 

A patient with diabetes, regardless of the type, must be aware of the chronicity of the disease, the importance of individual pharmacotherapy planned according to the type of diabetes, but also be aware that the course and progression of the disease, optimization of glycemia and severity of symptoms depend on his/her decisions and individual health behavior [16,17,18,19,20,21].

All recommendations adopted by diabetes associations cite lifestyle as the cause of diabetes, and lifestyle modification as a method of non-pharmacological treatment and therefore indicate therapeutic education as a systematic, multi-track and integrated approach to promote self-care practices among diabetic patients to prevent any long-term complications [11,12,13].

A diabetic patient prepared for self-care is a patient who has a certain amount of knowledge and specific skills. In terms of knowledge, it is necessary to understand the nature of the disease, its pathophysiology, metabolic changes, symptoms, state of hyper and hypoglycemia, as well as pharmacological and non-pharmacological methods of treatment. A patient with diabetes should learn how to monitor the symptoms, using a glucometer, administering insulin (pen, insulin pump), using a blood pressure measuring device, performing a urine strip test for ketone bodies and adjusting their menu to daily requirements [6]. In the event of the deterioration of the patient’s functional state, they will require support from a trained family and a prepared caregiver. The patient should know the principles of coordinated care for people with diabetes (primary health care and specialist care).

Preparation for self-care is one of the factors that determine the functioning of a chronically ill person, as it affects the patient’s ability to face the disease and to deal with any new or difficult situations arising from it.

Preparing a diabetic patient for self-care is the domain of a diabetes specialist nurse who is qualified and competent to prepare and conduct individualized educational programs for the patient and possibly their family/caregiver.

It is an important trend in modern nursing care, based on the expectation that the patient will take responsibility for themselves, for the results of treatment, for adherence to therapeutic recommendations, and for modifying their own lifestyle. The disease and the resulting limited efficiency might cause the patient to develop a kind of inability to be provided with the full range of care or deficits in this regard. The multifactorial determinants of diabetes, its type, duration, age of the patient, and their mental and physical condition, as well as broadly understood environmental conditions significantly shape the course of therapy [16,17,18].

In the current system of care for diabetic patients, the role of a physician is focused on diagnostics and treatment. The nurse takes part in patient care at every stage of the disease, and her activity-therapeutic education is intensive and parallel to diagnostic and therapeutic activities [12].

The aim of the study was to analyze the main aspects of preparing patients with diagnosed diabetes type I and type II for self-care. The analysis of the aspects of preparing the patient for self-care takes into account specific knowledge and particular skills, motivating the patient to change their lifestyle, to give up anti-health behaviors, and encouraging them to follow therapeutic recommendations, including dietary recommendations.

## 2. Material and Methods

### 2.1. Subjects

The study group consisted of 190 people diagnosed with type 1 and type 2 diabetes, including 101 women and 89 men. The presented results relate to the preliminary study, which is a part of a wider, prospective study. The entire project is planned for a period of 4 years with repeated research every 2 years.

The study was carried out at the Multispecialty Health Center “GRYF-MED” at 46 Wojska Polskiego Street in Bydgoszcz and at the Endocrinology and Diabetology Clinic of the Dr Antoni Jurasz University Hospital no.1 in Bydgoszcz at 9 Marii Skłodowskiej-Curie Street in Bydgoszcz.

Inclusion criteria for the study were as follows:Clinically diagnosed type 1 or type 2 diabetes;Age of 18 years or older;Able to fill in the questionnaire on their own;Consent to voluntary participation.

The study was conducted with the use of an anonymous, self-designed questionnaire containing 50 questions. The first part consisted of 27 questions of sociodemographic data and basic diabetes-related data. The second part of the questionnaire included 23 questions about lifestyle, treatment methods, and adherence.

Primary data was collected from participants via a self-reported questionnaire.

### 2.2. Study Protocol

The study was carried out in accordance with the Declaration of Helsinki. The protocol of the study was approved by the Bioethics Committee of the Nicolaus Copernicus University Collegium Medicum in Bydgoszcz, Poland (consent no. KB 362/2017). The survey was anonymous. 

### 2.3. Statistical Analysis

Statistical analysis was performed using the Statistica v. 10 software (StatSoft, Poland, Kraków) and the Excel spreadsheet Basic data was presented using quantity and value statistics (n/%). The correlation between the variables (age groups, education, duration of diabetes) was calculated using the Spearman’s correlation coefficient. The non-parametric Mann–Whitney U test was used to assess the differences in one trait between the two groups (gender, coexisting diseases, type of diabetes). The Kruskal–Wallis test (professional status, frequency of blood glucose measurements) was also used. Data are presented as mean ± standard deviation (SD). The level of statistical significance was *p* <0.05.

The analyses were presented descriptively for the entire group of patients diagnosed with diabetes. Demographic and behavioral data such as glycemic measurements and observations of physical symptoms were analyzed, but due to the lack of statistical significance, in-depth analyses were not performed, distinguishing between clinical groups (type I diabetes and type II diabetes). Only where the division into type I and type II diabetes was applied was the frequency of blood glucose measurements and the self-assessment of preparation for self-care in diabetes.

## 3. Results

Table 1 presents the general characteristics of the study group, which included 101 women and 89 men. The average age of the respondents was 42.2 ± 13.4 years. The oldest respondent was 72 years old, and the youngest was 18 years old. Most of the respondents 70.5% were city residents. Education was completed at the secondary level by 36.8% respondents, vocational 31.1%, higher 22.6% and primary 9.5. More than half of the respondents (63.7%) were economically active, 14.2% received retirement or disability benefits, 9.5% were students and 12.6% were unemployed. Every fifth respondent (20.5%) lived alone. Among the respondents, 72.6% were patients with type 2 diabetes. The mean time from diagnosis of diabetes was 7.7 ± 5.7 years. In the study group, 45.8% patients declared the presence of comorbidities. When answering the question about the type of comorbidities, the respondents selected several statements; 122 responses were recorded from 87 people in total. Most respondents (36.9%) indicated the presence of arterial hypertension. Further, the respondents indicated asthma (13.9%), obesity (8.2%), hypothyroidism (5.7%) and rheumatism (4.1%).

The treatment applied in diabetes is presented in Figure 1.

It is extremely important for patients with diabetes to monitor their glucose levels at home. Among the respondents, 23.2% did not control their glucose levels at home, while the vast majority (76.8%) declared that they self-measured their glucose levels. Every fifth respondent (21.1%) indicated the need to use the help of another person while measuring blood glucose. A total of 65.8% of the respondents reported that relatives assisted them in measuring glycemia. The respondents most often measured their glucose levels once a day (33.6%) or 3 times a day (26.7%), and only a slightly smaller group did so twice a day (26.0%) (Table 2).

The frequency of glycemic measurements was adopted by the researchers as an indicator of the patient’s self-observation capacity, interest in the stability of the glycemic state, and interest in his or her health condition. The age of the patients, the duration of diabetes treatment, the occurrence of hypoglycemia or hyperglycemia in the last 6 months were not significantly correlate with the frequency of blood glucose measurements performed by the patients (*p* > 0.05). Among the respondents, the professional status did not have a statistically significant correlation (*p* > 0.05) with the frequency of blood glucose measurements. The incidence of comorbidities among patients did not translate into the frequency of glycemic measurements (*p* > 0.05).

There was a statistically significant difference between the groups of patients depending on the type of diabetes, in terms of the frequency of blood glucose measurements (*p* = 0.004). To a greater extent, glycemic controls were performed by patients with type I diabetes.

The measurements of glycemia and arterial pressure can be used for keeping a self-care diary, in which the said values are regularly recorded. Managing a self-care diary was reported by 64.7% of the respondents.

For diabetic patients, an important skill is to observe the organism for potential glycemic disorders (hypoglycemia, hyperglycemia) and to react accordingly. The level of the symptoms knowledge and the methods of hypoglycemia management was declared by 64.8% of the participants. At the same time, knowledge related to hyperglycemia symptoms was declared by 65.8% of the patients, and of methods for its management—63.7%. Despite being aware of the symptoms of glycemic disorders, during the 6 months before the study, 4.7% of the respondents experienced acute hyperglycemia, and 6.3% experienced acute hypoglycemia. Most patients (70.7%) declared their participation in education programs, preparing chronically ill patients to live with diabetes, organized and managed by competent healthcare professionals: diabetes doctors and nurses, family doctors and nurses, as well as dieticians. Moreover, every third respondent (29.3%) learned about diabetes from other sources, such as the Internet, books journals, television, and friends. The majority of the patients (75.8%) declared that they had sufficient knowledge and skills needed to manage self-care in diabetes. Despite such a high number of respondents declaring satisfactory knowledge and skills, negative health behaviors were still displayed by the patients. Smoking cigarettes applied to 43.2%, while alcohol consumption, ranging from mild to severe, was declared by as many as 86.8% of the respondents. Half of the study population (53.5%) reported using stimulants other than cigarettes or alcohol. When asked about the type of other stimulants used, the respondents marked several statements: 121 answers were given by a total of 107 patients (the rest did not indicate any stimulants). Among the respondents, 82.6% declared the consumption of coffee, and 16.5% declared the consumption of energy drinks. 

What is more, the respondents reported not being engaged in sufficient physical activity. As many as 52.1% failed to take exercise lasting over 30 min at least three times a week. Three patients (1.6%) declared being engaged in at least 150 min of physical activity per week. The remaining respondents (46.3%) undertook physical activity sporadically.

Most of the patients (73.7%) declared suffering from stress, while every third respondent (31.5%) reported that they were able to cope with stressful situations. 

The respondents were asked to self-assess their health status. As many as 75.2% of the patients described their condition as very good or good (Figure 2). 

For patients with a chronic disease such as diabetes, the key issue is pharmacotherapy, in which they should actively participate. Pharmacotherapy is individually prepared for each patient, taking into account their clinical condition, capacity, fitness and physical activity, daily efficiency and functional activity, as well as living conditions. Independent participation in therapy, i.e., taking hypoglycemic drugs or insulin, was declared by 63.7% of the respondents. Among patients who were dependent in their pharmacotherapy, 7.9% required constant assistance of other people. Among 76 patients treated with subcutaneous insulin, 76.3% reported self-preparation and self-injection. All patients using insulin pumps (39) were independent in the use of the equipment thanks to instructions with which they had been provided. In the analysis of the occurrence of hypoglycemia and hyperglycemia symptoms in the last 6 months in the above-mentioned four groups of patients differing in the type of therapy in diabetes, no statistically significant difference was found (*p* = 0.437).

As mentioned above, 45.8% of the respondents were diagnosed with other diseases accompanying diabetes (multi-morbidity) and were subject to combination therapy, including complex treatment regimens. In this group, every fourth patient (27.4%) reported experiencing various ailments associated with taking medications other than hypoglycemic drugs, and every second patient 51.9% reported that they stopped taking medications when ailments occurred.

The duration of treatment in diabetes mellitus was not statistically significantly correlated with the results of hypoglycemia and hyperglycemia symptoms during the last 6 months.

Among 75.8% of the respondents declaring a sufficient level of preparation for self-care in diabetes, 33.3% declared a willingness to explore their knowledge about the disease. On the other hand, among 24.2% of people indicating the lack of preparation for self-care in diabetes, only 32.6% declared the need to explore their knowledge about the disease.

It was shown that the level of preparation for self-care in diabetes increased significantly with the patient’s age (*p* = 0.001).

Education did not determine the patient’s level of preparation for self-care in diabetes.

There was a statistically significant difference between the group of patients with type I diabetes and the group of patients with type II diabetes in terms of preparation for self-care in the disease (*p* = 0.001). The declaration of sufficient preparation for self-care in diabetes was indicated to a greater extent by patients with type I diabetes (Table 3).

## 4. Discussion

The conclusion that comes to mind from the preliminary examination is that sociodemographic data do not determine the level of the patient’s preparation for self-care. The subjective assessment of the level of preparation for self-care is higher in the group of patients with type I diabetes. Health behaviors do not differ among the respondents. A patient with type II diabetes is diagnosed late and is therefore burdened with an increased risk of complications. In our own study, as many as 21% of patients with type 2 diabetes are insufficiently prepared for self-care in diabetes and, at the same time, do not express the need to supplement their knowledge about the disease. Researchers regard this result as disturbing. It indicates the need for active and intensive educational work of a diabetic nurse with a patient to reduce the risk of developing vascular complications stroke or myocardial infarction. Diabetes complications also include nerve damage, i.e., diabetic neuropathies. Diabetic foot syndrome is also a common complication [12,22]. For example, in Poland in 2020, the total percentage of patients diagnosed with diabetes for those starting renal replacement therapy using hemodialysis was 32% [23]. Based on OECD data published in the Health at a Glance 2019 report, the age- and gender-standardized number of major lower limb amputations in adults with diabetes per 100,000 people vary between countries. Of the 31 countries included in the list, Poland had the tenth highest value of the rate of major lower limb amputations in adults with diabetes per 100,000 population [2].

The nurse plays an important role in managing self-care and preparing the patient for self-care. 

A specialist nurse has competences, as well as professional preparation to fulfil the educational role, management and support of the patient and their family. In their work, they can use modern technical possibilities (media, communicators, data transfer, and monitoring systems).

The nurse is predestined to the role of a care coordinator for a diabetic patient who is a subject of care in the primary health care system and specialist care.

Due to the course and duration of the disease, as well as the impact of the patient’s health behavior on the risk of late complications and treatment costs, the International Diabetes Federation has developed guidelines for patient care and prevention of diabetes [5], which are consistent with the standpoint of the Polish Diabetes Society [6]. It emphasizes that the preparation of the patient for self-care and self-management takes place through education that is individualized, focused on the patient, and taking into account their clinical condition, perceptual abilities and environmental conditions. Education of a diabetic patient should be comprehensive and carried out by a competent team. It should also be combined with behavioral therapy, which aims to correct health behaviors. An inseparable element of education is psychological support provided to the patient, focused on encouraging acceptance of a chronic disease, strengthening the motivation for optimal management in the therapy process, and shaping the patient’s sense of influence on the course of the disease, while at the same time ensuring open communication with the therapeutic team. Education in the treatment of diabetic patients is so important that it is recommended to involve not only the patients themselves, but also their families and caregivers [6].

Referring to the description of a patient characterized as prepared for self-care, it is difficult to talk about full co-responsibility and active participation of the patient in the therapeutic process, since as many as 23.2% of the patients included in the present study did not control their glucose levels at home with a glucometer, and every fifth respondent (21.1%) reported using the help of another person in taking this measurement. In this study, 64.7% of the respondents declared keeping a self-care diary. This result is consistent with the data presented in the report of the Coalition for Fighting Diabetes of 2017, where 58% of respondents managed a self-care diary [24].

In the present study, self-preparation and administration of oral hypoglycemic drugs or insulin was reported by 63.7% of patients. A potential lack of independence of a patient indicates the need to provide them with physical rehabilitation, to select medical equipment adapted to their psychophysical abilities, and—in many cases—to use the support of relatives. 

For patients with a chronic disease such as diabetes, it is crucial to learn about their internal resources: the sense of agency, of being able to manage the disease, of independence, and of security. These contribute to the patient’s independence and the ability to manage self-care. With professional psychological assistance, it is easier for the patient to discover their individual internal resources [25,26].

In the treatment of people with diabetes, it is recommended to strive for modification and transition from a sedentary to a more active lifestyle with the use of all forms of activity [5,6]. Physical activity should be undertaken regularly. Before and after exercise, it is advisable to determine blood glucose levels corrected by taking an additional portion of carbohydrates [27,28]. Although, regular physical activity for at least 30 min is conducive to reducing the risk of cardiovascular diseases and is an important aspect of non-pharmacological treatment of diabetes, the present study showed that 52.1% of the patients did not consider it as a permanent element in their lifestyle. Only three patients (1.6%) declared systematic physical activity of about 150 min a week. In the National Health Test of Poles in 2020, among 401,195 respondents, the largest group of people (39%) was engaged in physical activity for up to 30 min a day [29].

Chronically ill patients require systematic therapy, following indications and recommendations of their physician, as well as compliance with treatment procedures in order to minimize the effects of the disease. Failure to comply with the principles of long-term therapy reduces the effectiveness of treatment, which most often leads to the patient discontinuing the therapy and, consequently, causes a greater number of complications and increases mortality. Complications reduce the patient’s quality of life and burden the health care system [3,30].

The authors’ own study showed that patients with type I diabetes declared, to a greater extent, more sufficient preparation for self-care in diabetes than patients with type II diabetes. It was confirmed by other readers [24].

The standard for the management of diabetic patients adopted by the Polish Diabetes Society provides for comprehensive patient care and is based on the assumption that compliance with this standard prevents the progression of the disease and the development of serious complications. The standard defines the educational role of the nurse in preparing the patient for self-care. Nurses working at all levels of diabetes care can contribute to organizing and delivering high-quality care for patients with diabetes. They play an important role in shaping and supporting the patient’s responsibility for their own health through face-to-face consultations, counseling, or the provision of structured diabetes education and self-control plans [8,31,32,33,34].

The presented results relate to the preliminary examination. The project had some limitations. The authors did not analyze the results of glycemic tests in the study due to the different types of measuring devices, both in patients and in health care facilities. Looking at the results of the research, it becomes clear that the study should be extended to include psychological elements and the study of motivation. It is difficult to decide on generalizations when the study group is not very large. It should be emphasized the important aspect in the work of preparing the patient for self-care is to establish a positive, constructive nurse–patient relationship. The patient should perceive the nurse as a competent and professional person.

## 5. Summary

Patients with a chronic disease such as diabetes require coordinated care and individualized therapeutic education in order to be prepared for self-care and self-management, thus reducing the risk of complications. Individual assessment of the degree of the patient’s knowledge and skills is of great importance for the clinician/therapist while preparing the patient for self-care and self-control. These include, among other things, simplification of therapy regimens, education of the patient and their family, psychotherapy, crisis intervention and refresher courses.

## 6. Conclusions

We have demonstrated that despite receiving therapeutic education, the study participants diagnosed with diabetes still show deficiencies in terms of awareness of proper health behaviors. Objective results showed that the patients had insufficient knowledge and skills in terms of self-care and self-observation, blood glucose and blood pressure measurements, physical activity, diet therapy and adherence to pharmacotherapy recommendations. Patients with type I diabetes showed a greater level of preparation for self-care in the disease.

Despite the good general preparation for self-care declared by patients with diabetes, these patients require further systematic, individual educational activities. Diabetic patients and their families should be included in the education program with the aim of increasing their awareness of diabetes complications.

## Figures and Tables

**Figure 1 ijerph-19-11365-f001:**
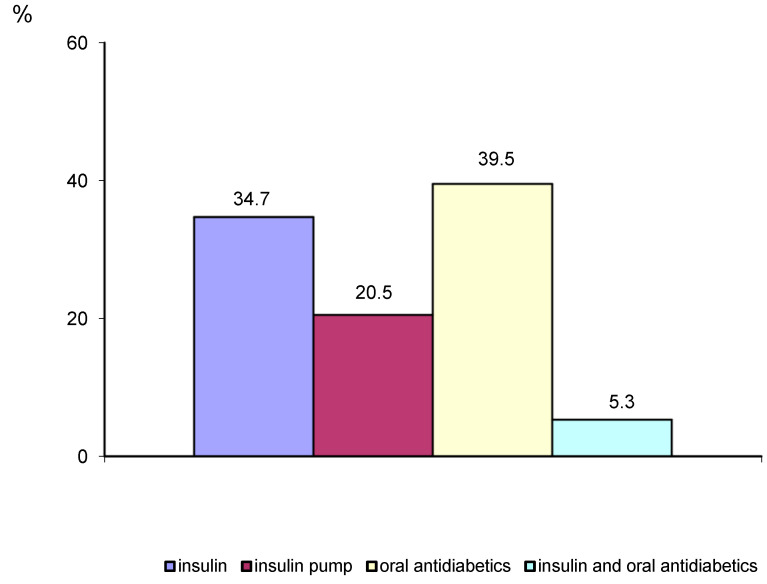
Applied treatment of diabetes mellitus.

**Figure 2 ijerph-19-11365-f002:**
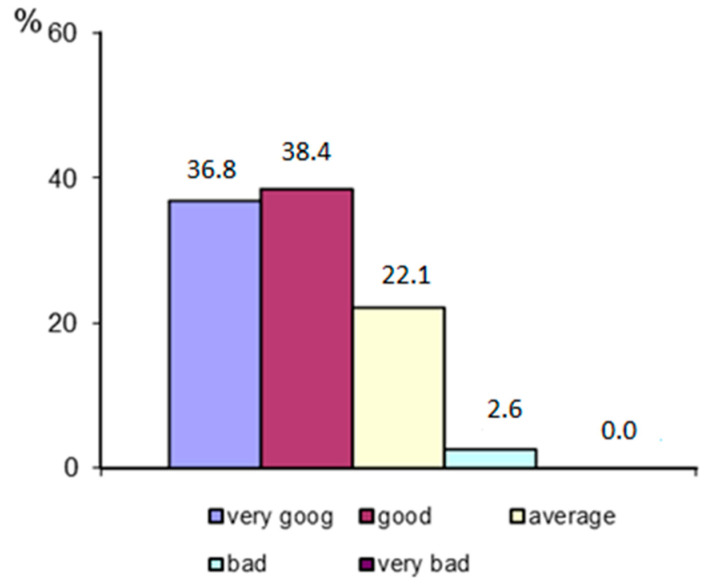
The respondents’ self-assessment of health status.

**Table 1 ijerph-19-11365-t001:** The general characteristics of the study population.

Demographic Characteristics of Respondents	Group Size n = 190 (%)
Sex	
male	89 (46.8%)
female	101 (53.2%)
Age (in years)	
up to 30	49 (25.8%)
31–40	36 (18.9%)
41–50	44 (23.2%)
over 50	61 (32.1%)
Place of residence	
country	56 (29.5%)
city	134 (70.5%)
Education	
primary	18 (9.5%)
vocational	59 (31.1%)
secondary	70 (36.8%)
higher	43 (22.6%)
Business activity	
unemployment	24 (12.6%)
full-time job	121 (63.7%)
disability pension/retirement	27 (14.2%)
pupil/student	18 (9.5%)
Living	
with family	151 (79.5%)
alone	39 (20.5%)
Co-existing diseases	
hypertension	45
asthma	17
obesity	10
hypothyroidism	7
rheumatism	5
Type of diabetes	
type 1	52 (27.4%)
type 2	138 (72.6%)

**Table 2 ijerph-19-11365-t002:** Frequency of measuring blood glucose levels.

Frequency	No.	%
5× day	1	0.7
4× day	10	6.8
3× day	39	26.7
2× day	38	26.0
1× day	49	33.6
every 2 days	3	2.1
1× week	6	4.1
Total	146	100.0

**Table 3 ijerph-19-11365-t003:** Declaration of sufficient preparation for self-care in diabetes.

Type Diabetes	Type I		Type II	
Answers	No.	%	No.	%
yes and it is sufficient	35	67.3	61	44.2
yes, but I would like to know more about diabetes	13	25.0	35	25.4
no, and I don’t need such knowledge	2	3.8	29	21.0
no, but I would like to know more about diabetes	2	3.8	13	9.4
Total	52	100.0	138	100.0

## Data Availability

The data presented in this study are available on request from the corresponding author.

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
