# Peer review of "Main Aspects of Preparing Diabetic Patients in Poland for Self-Care"

_ijerph, 2022, doi:10.3390/ijerph191811365_

Round 1
Reviewer 1 Report
The paper reports the result of an anonimous questionare on disease awareness and principles of self care delivered to a relatively small population of diabetic patients.
The paper is to a large extent unfocused and collect informations on many aspects of clinical and behavioral attitude of diabetic patients.
The major limitation of the study is the data analysis that merge together both type 1 and type 2 diabetic patients.
No correlations are made with clinical data (duration of the disease, blood glucose control, complications), same albeit limited data are provided on comorbidities but are not propperly analyzed.
Reviewer 2 Report
This study investigated factors that could determine selfcare management for patients with diabetes. It collected primary data from patients in Poland. Findings showed that 22% of patients still needs assistance for measuring glucose level properly and about half of patients does not maintain healthy lifestyle. It may cause negative consequences such as complications and other comorbidities. A strength of this study was primary data with 50 questions containing demographics, lifestyle and behavior for diabetes management.
Overall, the study needs major revisions and the writing should be revised in professional language. Based on review, my decision is major revision with substantial edits in language. The below is details of suggestions.
Title: Title should be precise and clear to highlight your research. “Some” is not professional term to be used in the title.
Introduction and aim:
Overall, the introduction should give in-depth information regarding self-care management led by nurses. For instance, it needs to describe more about how current nurses help and educate patients for self-care and how this process is significant to prevent adverse outcomes for patients.
Also personal responsibility sounds vague. Should define what personal responsibility and how nurses (if you focus on nurses) can boost personal responsibility for patients.
Page 2, line 60-63 and line 66-67: don’t understand what those sentences mean. Revise the language.
Aim pointed out ‘selected aspects’ of preparing patients. I wonder that preparation represents education or give guideline when patient is diagnosed. Should clarify this preparation.
In addition, selected aspects are not defined in previous paragraph. It should be identified.
Methods
Sampling methods should be described. Also in-depth explanation of development survey is needed. What are the domains in the survey such as demographics and behaviors?
Page 2, line 87-88: revise the language to “a primary data was collected from participants via self-reported questionnaire.”
Should report
Statistical analysis has critical flaws. First, authors used the term “qualitative data” but it is not correct. Numeric values collected from survey and coded into the program are quantitative data not qualitative data. Should differentiate qualitative and quantitative data. Chi-square test cannot be used for qualitative data.
Second, the correlation cannot be measured by chi-square test. Chi-square test is a way of identifying significant difference between groups with categorical variables.
Third, the study did not clarify comparison groups. Should clarify who you are comparing with.
Results
Results should report key findings from the table. Many redundant information is described. I would recommend to report only percentage not sample size. It confuses readers.
Page 3, line 108: numbers don’t match to the table. Double check the numbers throughout the results.
There is no p-value reported for comparison. As aforementioned, comparison is not clearly stated in methods nor in results.
Clarify comparison groups and report this comparison via table.
Need one more table that displays the differences of behaviors with significance.
Discussion
Discussion is confusing to highlight key findings of the study. The first paragraph does not summarize findings and deliver indication. Especially, if authors want to emphasize the role of nurses for self-care management, the contribution and role of nurses should have been addressed in relation to findings. Current discussion is not supportive to highlight the need of resource allocation and guidance.
The study does not discuss limitations of the study and the generalizability
Author Response
Dear Reviewers,
thank you very much for the valuable, constructive comments made in the review of the article.

Round 2
Reviewer 2 Report
The revised study addressed some of my comments. However the authors still need to respond to the remaining of my comments and address them in the manuscript for improving the quality of study with clear topic. Thus, I would recommend major revision again.
Introduction
1) The first comment is not addressed. Initial comment: “Overall, the introduction should give in-depth information regarding self-care management led by nurses. For instance, it needs to describe more about how current nurses help and educate patients for self-care and how this process is significant to prevent adverse outcomes for patients.” If the study highlighted the role of nurse in self-care management for patients, the significant benefit of nurse-led intervention such as education or discharge plan should be explained with evidence.
2) Page 2, line 58-69. This paragraph should explicate Dorothea Orem’s theory and how this theory can be applied to care coordination with patient and nurse for self-care preparation. Current paragraph just described the treatment regimens for type 1 and type 2 diabetes. It is not relevant to care coordination led by nurse.
Methods
1) The first comment is not addressed, “Sampling methods should be described. Also in-depth explanation of development survey is needed. What are the domains in the survey such as demographics and behaviors?” Should describe recruitment on site.
2) The last comment in methods is not addressed, “Third, the study did not clarify comparison groups. Should clarify who you are comparing with.” Still don’t know which group is comparting to which group. Results reported the comparison between type 1 and type 2 diabetes. Treatment regimen and clinical guideline should be different for these two diabetes as the authors stated in introduction. If these two groups are comparison group, should justify why you compare these two groups in Introduction and methods.
3) The correlation was conducted among age, education and duration of T2D. It is already proved in many literature via regression model. And the interpretation of this correlation is not correct. Correlation does not determine causal relationship.
4) Page 3, line 106: typo-repeated
Results
1) The following two comments are not addressed. “Clarify comparison groups and report this comparison via table.” And “Need one more table that displays the differences of behaviors with significance.”
2) Page 7, line 236-237. Interpretation of correlation is wrong.
3) Page 7, line 239-241. Report numbers in table to show how those two groups are different with significant value.
Discussion
1) The first comment is not addressed. “Discussion is confusing to highlight key findings of the study. The first paragraph does not summarize findings and deliver indication. Especially, if authors want to emphasize the role of nurses for self-care management, the contribution and role of nurses should have been addressed in relation to findings. Current discussion is not supportive to highlight the need of resource allocation and guidance.”
2) Page 7, line 243-258: New paragraph is confusing to find key findings of the study. Should revise the language in clear and precise way.
3) Page 8, line 259: typo in the beginning of the paragraph.
Author Response
Dear Reviewer,
thank you very much for the valuable, constructive comments made in the review of the article.
Introduction
Ad. 1. The introduction was corrected.
Page 2, line 59-110
Ad. 2. Page 2, line 58-69 were corrected.
Methods
Ad. 1. The entire survey included 50 questions. The first part of the questionnaire contained 27 questions. Among them were:
- Demographic data analyzed in the study (age, gender, place of residence, education, professional activity).
- Diabetes data included type of diabetes, duration of disease, glycemic control (frequency, self-measurement, interpretation of results).
- Diabetes education data (who led, sources, methods).
- Data on therapy education (type of therapy: insulin, oral hypoglycemic drugs, pump), observation of symptoms: hypoglycemia, hyperglycemia, taking actions in the state of hypoglycemia, hyperglycemia.
The second part of the questionnaire - 23 questions, concerned health-related behaviors, including:
- Nutrition (number and composition of meals, self-assessment of the level of preparation in terms of nutrition in diabetes).
- Concerning smoking.
- Concerning alcohol consumption.
- Concerning physical activity.
- Concerning exposure to stress and coping with stressful situations.
The remainder of the section contained questions about adherence to treatment. There was also a question about control from specialist clinics. An important question was the subjective assessment of the level of preparation for self-care in diabetes.
The inclusion criteria for the study group was supplemented with a questionnaire able to complete independently.
Ad.2.
Two groups were not analyzed in the study. A group of patients with type I and type II diabetes was analyzed together. The only point where patients with type I and type II diabetes were analyzed separately was the question about the subjective assessment of preparation for self-care in diabetes and an examination of the frequency of glycemic measurements.
The reason why the groups were not separated in the preliminary study was that in the group of patients with type II diabetes included patients who were also treated with insulin (three treatment methods: dietary treatment, oral hypoglycemic drugs, insulin injections). The authors of the study wanted to present the main directions of education in a group of diabetic patients led by a diabetes nurse.
Ad.3.
The analyzes were presented descriptively for the entire group of patients diagnosed with diabetes. Demographic data and behavioral data such as glycemic measurements and observations of physical symptoms were analyzed, but due to the lack of statistical significance, in-depth analyzes were not performed, distinguishing between clinical groups (type I diabetes and type II diabetes). Only where the division into type I and type II diabetes was applied was the study of the frequency of glycemic measurements and the self-assessment of preparation for self-care in diabetes.
The frequency of glycemic measurement was adopted by the researchers as an indicator of the patient's self-observation capacity, interest in the stability of the glycemic state, and interest in his or her health condition. Patients' age, duration of diabetes treatment, and occurrence of hypoglycemia or hyperglycemia in the last 6 months did not significantly correlate with the frequency of patient blood glucose measurements (p> 0.05). Among the respondents, the professional status did not have a statistically significant correlation (p> 0.05) with the frequency of blood glucose measurements. The presence of comorbidities among patients did not translate into the frequency of glycemic measurements (p> 0.05).
Ad. 4. Page 3, line 106: was corrected
Results
Ad.1.
The analyzes were presented descriptively for the entire group of patients diagnosed with diabetes. Demographic data and behavioral data such as glycemic measurements and observations of physical symptoms were analyzed, but due to the lack of statistical significance, in-depth analyzes were not performed, distinguishing between clinical groups (type I diabetes and type II diabetes). Only where the division into type I and type II diabetes was applied was the study of the frequency of glycemic measurements and the self-assessment of preparation for self-care in diabetes.
Ad. 2.
Page 7, line 236-237. Interpretation of correlation was corrected.
Education did not determine the patient's level of preparation for self-care in diabetes (p> 0.05).
Ad. 3. Page 7, line 239-241. Table 3 was inserted.
Discussion
Ad. 1. Discussion was corrected.
Ad. 2. Page 7, line 243-258: New paragraph was corrected
Ad. 3. Page 8, line 259 was corrected
